# Weight Status Is Related to Health-Related Physical Fitness and Physical Activity but Not to Sedentary Behaviour in Children

**DOI:** 10.3390/ijerph17124518

**Published:** 2020-06-23

**Authors:** José Francisco López-Gil, Javier Brazo-Sayavera, Juan Luis Yuste Lucas, Fernando Renato Cavichiolli

**Affiliations:** 1Departamento de Actividad Física y Deporte, Facultad de Ciencias del Deporte, Universidad de Murcia (UM), 30720 San Javier, Region of Murcia, Spain; 2Polo de Desarrollo Universitario EFISAL, Centro Universitario Regional Noreste, Universidad de la República (UDELAR), 40000 Rivera, Uruguay; jbsayavera@cur.edu.uy; 3Departamento de Expresión Plástica, Musical y Dinámica, Facultad de Educación, Universidad de Murcia (UM), 30100 Espinardo, Region of Murcia, Spain; jlyuste@um.es; 4Departamento de Educação Física, Universidade Federal do Paraná (UFPR), 81540-410 Curitiba, Paraná, Brazil; cavicca@hotmail.com

**Keywords:** obesity, adiposity, waist circumference, screen time, children, sport activities

## Abstract

*Purpose:* The aim of this research was to describe, examine, and compare the level of physical fitness, physical activity, and sedentary behaviour in pupils aged 6–13 in the Region of Murcia, Spain, in accordance with weight status. *Methods:* A total of 370 children (166 girls and 204 boys) aged 6–13 (M = 8.7; DT = 1.8) from the Region of Murcia participated in this descriptive and cross-sectional study. Some anthropometric parameters such as body mass index, waist circumference, as well as skinfold measurements were determined. ALPHA-FIT Test Battery was used to evaluate physical fitness. Krece Plus Short Test was used to measure physical activity level and sedentary behaviour. *Results:* 52.4% of the children presented excess weight (according to the World Health Organization growth references). Regarding boys, statistically significant differences were found for cardiorespiratory fitness (*p* < 0.001), relative handgrip strength (*p* < 0.001), lower muscular strength (*p* < 0.001), speed-agility (*p* < 0.001), as well as sport activities hours (*p* = 0.001) among the three weight status groups (normal weight, overweight, and obesity). As for girls, statistically significant differences were found for cardiorespiratory fitness (*p* = 0.004), relative handgrip strength (*p* < 0.001), lower muscular strength (*p* < 0.001), sport activities hours (*p* = 0.005), as well as physical activity level (assessed by Krece Plus Test) (*p* = 0.017). A negative statistically significant correlation was found between body mass index and cardiorespiratory fitness (*rho* = −0.389), lower muscular strength, (*rho* = −0.340), and relative handgrip strength (*rho* = −0.547). At the same time, a positive statistically significant relationship between body mass index and the time spent in speed-agility (*rho* = 0.263) was shown. Regarding waist circumference and body fat percentage, similar relationships were identified. Moreover, a greater probability of having higher cardiorespiratory fitness (OR = 1.58; CI_95%_ = 1.38–1.82), relative handgrip strength (OR = 1.25; CI_95%_ = 1.19–1.31), more hours of sport activities (OR = 1.40; CI_95%_ = 1.19–1.66), and physical activity level (assessed by Krece Plus Test) (OR = 1.23; CI_95%_ = 1.07–1.42) was noted in the normal weight group. *Conclusions:* Children that presented normal weight achieved higher results for health-related physical fitness and physical activity than those with excess weight; this was, however, not found to be the case for sedentary behaviour. The authors emphasise the need for changes in public policies and school-based intervention programmes to develop higher levels of both PF and PA in overweight and obese children.

## 1. Introduction

The global prevalence of overweight and obesity in children and adolescents is alarmingly high [1]. In Spain, the prevalence of obesity in children between 6–9 years old in 2015 was 23.2% and 18.1% for overweight and obesity, respectively; 41.3% for excess weight [2]. The presence of excess weight (understood as the sum of overweight or obesity) in childhood and adolescence increases the risk of morbidity and mortality in adulthood [3], and has a great influence on risk factors for cardiovascular disease and the development of atherosclerosis [4]. Likewise, childhood obesity increases the likelihood of having higher risk parameters (cholesterol, triglycerides, blood pressure level, etc.) for cardiovascular disease in adulthood [5].

Physical fitness (PF) is recognised as an independent predictor of cardiovascular disease and adulthood mortality [6,7]. Poor cardiorespiratory fitness (CRF) and muscular fitness levels are linked to cardiovascular disease risk factors such as adiposity and some metabolic risk parameters in childhood [8]. Moreover, some studies have reported a negative association between excess body fat and some parameters of health-related PF, such as CRF [9,10] or muscular fitness [11,12].

On the other hand, obesity has been associated with different behaviours connected with low levels of physical activity (PA) [13] and sedentary habits [14]. Recent reports have suggested that PA is the most variable component of total energy expenditure; with only 30–40% of young people physically active (assessed by self-report) according to public health recommendations for youths aged 5–19 (a minimum of 60 min daily of moderate-to-vigorous PA) [15,16]. Likewise, the report card on PA indicators from the Global Matrix 3.0 described fairly low levels in PA level of children in Spain [17]; therefore, there is a need for local information on this indicator in order to establish effective public policies to increase population levels of PA, because if the current trend continues, the 2025 global PA goal (a 10% relative reduction in insufficient PA) will not be achieved [18]. In terms of sedentary behaviour (SB), a high number of screen time hours could displace the time spent in PA, increase energy intake from eating while viewing, and/or reduce sleep [14]. Unfortunately, low adherence to PA in Spanish children and adolescents [16], as well as a lack of compliance with the international recommendations on screen time (<120 min per day) [19] was recently reported.

In Spain, studies have been performed to examine the relationship between weight status and PA level [20,21], PF [22,23], or even SB [20,24]—showing a negative relationship between most of them. Despite this fact, the limitation of periodic data on both PA and PF in Spain, as well as the lack of local representativeness (since Spain has broad cultural differences among regions) has been pointed out [17]. Likewise, other studies carried out in the Region of Murcia were focused principally on the relationship of PF and weight status [25,26,27], although most of them only determined BMI [25,26], while the inclusion of other anthropometric indicators is specially recommended [28]. In addition, there is the need for continuous surveillance and provision of data from different regions to identify cultural differences in the same country and guide decisions on prevention efforts [29]. In addition, enhancements in the processes, tools, and outcomes for youth surveillance will improve the monitoring of physical activity indicators [16].

On the other hand, it must not be forgotten that the process of cardiovascular disease begins in childhood, and associated risk factors, including insufficiency of PA and obesity, track through adolescence into adulthood, resulting in an increased risk of premature mortality [30]. In this manner, changes in public policies and intervention programs focused on weight reduction and lifestyle improvement could be designed more efficiently in order to reduce the risk of children becoming obese as adults [31]. Hence, identifying children who are at high risk of adult obesity by measuring obesity in childhood and related factors could be a fundamental aspect; especially in the Region of Murcia, which has recently been pointed out as a region of Spain with the highest number of children with excess weight [32].

According to the evidence and the lack of local information in most parts of Spain, particularly in the Region of Murcia, the aim of this study was to describe, compare, and analyse the level of PF, PA, and SB in children aged 6–13 in the Region of Murcia, in accordance with weight status.

## 2. Materials and Methods

### 2.1. Study Design and Participants

A descriptive and cross-sectional study was conducted in six primary schools in the Region of Murcia (Spain). A total of 370 children (166 girls and 204 boys) aged 6 to 13 (M = 8.7; DT = 1.8) with similar socio-demographic features participated. In terms of age groups, 230 schoolchildren were between 6–9 (118 boys and 112 girls) years of age. In the age group of 10–13, 140 schoolchildren were found (86 boys and 54 girls). Non-probability sampling was the technique used in this study. All children were given the option to take part in the study.

In order to enrol a child in the study, parental/legal guardian permission was requested. Both parents and children were duly informed through a document explaining the aim of the study and the procedures involved. Students who were exempted from physical education classes were not included.

The present study was approved by the Bioethics Committee of the University of Murcia (ID 2218/2018). It was conducted following the Helsinki Declaration, respecting the human rights of the participants.

### 2.2. Procedures

#### 2.2.1. Anthropometric Measurements

In order to determine the participants’ height, a transportable height rod with a precision of 0.1 cm (Leicester Tanita HR 001, Tokyo, Japan) was used. Body weight was determined with a body composition analyser with precision of 0.1 kg (Tanita BC-545, Tokyo, Japan). Body Mass index (BMI) was computed from the ratio between body weight (kg) and the children’s height squared (m^2^). Besides, BMI z-score was computed using the age-specific and sex-specific thresholds provided by the World Health Organization (WHO) [33] in order to establish three different weight status categories: normal weight (median), overweight (>+1 standard deviation), and obesity (>+2 standard deviation). Waist circumference was measured with a precision of 0.1 cm at the intersection between the border of the iliac crest and the last rib, with a constant tension tape. Skinfold measurements were taken with a precision of 0.2 mm using pre-calibrated steel callipers (Holtain Ltd., Crosswell, Crymych, UK) at the biceps, triceps, subscapular, and iliac crest. Guidelines from the International Society for the Advancement of Kinanthropometry (ISAK) were considered in all these procedures. To calculate body density, the log of the sum of skinfolds was applied [34]. To calculate body fat from body density, the Siri formula was used [35] and fat-free mass was then determined by the subtraction between total body mass and body fat mass. 

#### 2.2.2. Physical Fitness

The ALPHA-FIT Test Battery included PF tests for the youth population [36], which were applied to evaluate the elements outlined below. To estimate CRF, a maximum incremental field test was performed (20 m shuttle run test). The children had to run between two lines 20 m apart, maintaining the rhythm of the acoustic signals given by a Bluetooth speaker. 8.50 km/h was the initial speed, which was increased by 0.5 km/h every minute, until it reached 18.0 km/h in the 20th min. Children were given instructions to run in a straight line and turn around when completing the itinerary between both lines and to maintain the rhythm indicated by the audible signals. During the course of the test, we encouraged the participants to run for as long as they could. The test was considered complete when the last subject failed twice to reach one of the lines in line with the audio. On the contrary, the test was over if the participant stopped due to exhaustion. These stages were converted to relative values of maximum oxygen consumption using Léger’s et al. [37] equations.

To measure upper body muscular fitness, we measured handgrip strength by means of a hand dynamometer with adjustable grip (TKK 5401 Grip D, Takei, Tokyo, Japan). To begin with, the children were given a brief explanation. We regulated the dynamometer to meet the child’s hand size as it had been formerly recommended [36]. This test was performed in the standing position, maintaining the elbow extended and the wrist in a neutral position. The subjects were told to “squeeze as hard as possible” and, for at least two seconds, apply maximal strength. We performed two attempts per hand, and the best result was recorded. In the analysis, the average of the best results achieved by each hand was used [36]. Besides, the handgrip score (kg) was calculated as the average of the left and right and then expressed per kilogram of body weight [36].

On the other hand, in order to calculate lower body muscular fitness, a standing broad jump was executed. Subjects were asked to stand with feet together behind the starting line. The subject had to jump forward as far as possible. The distance was measured from the take-off line to the landing point (back of the heels). Two attempts were performed and the best result was recorded (in cm) [36]. 

Speed-agility was measured with the 4 × 10 m Shuttle Run Test. The test required the participants to run back and forth between two lines (10 m of distance between them) as fast as possible. When the acoustic signal was sounded, the participants ran to the other line and took (the first time) or exchanges (the following times) a sponge that was formerly located behind the lines. The test ended when the participant crossed the finish line with one foot. Two attempts were made and the quickest result was recorded (in seconds) [38]. 

#### 2.2.3. Physical Activity and Sedentary Behaviour

The Krece Plus Short Test was validated for youths aged 4–14 in the EnKid Study [20] and was used to measure PA level and SB. This test is used to measure the usual PA level (0–10) of children according to the average amount of time (in hours) per day they watch TV or play video games, and the number of sport activities hours per week. Likewise, the children were asked about the number of hours that they usually spend doing sport activities: “How many hours do you spend on sports activities each week?”. Moreover, children were asked for the number of hours per usual day: “How many hours do you watch TV or play video games on average every day?”. All the children answered for themselves with the help of the assigned researcher.

#### 2.2.4. Statistical Analysis 

Frequencies and percentages (%) are presented for all qualitative variables and means (M) and standard deviation (SD) are described for all quantitative variables. Data normality was verified by Shapiro-Wilk (*n* ≤ 50) or Kolmogorov-Smirnov tests with Lilliefors correction, and the homogeneity of variances by the Levene test.

Consequently, the data were examined with the Kruskal-Wallis *H* test or one-way ANOVA for three-group comparisons (normal weight, overweight, and obesity), according to the normality assumption. Likewise, post hoc analyses were carried out in order to test differences between groups. In the case of variables without assumption of normality, the Mann-Whitney *U* test with Bonferroni correction was executed. Conversely, for normal variables, Tukey’s HSD or the Dunnett T3 test, depending on the homogeneity of the variances, was used. Effect size was estimated by Cohen’s *d* (0.20, small; 0.50, medium; and 0.8, large effect). The relationships between quantitative variables were also identified using Spearman’s rho (*ρ*). Besides, multinomial logistic regression was performed to predict the probability of getting different results according to the weight status. Finally, data analysis was performed using the software SPSS (IBM Corp, Armonk, New York, USA) for Windows (version 24.0). Statistical significance was denoted with a *p*-value ≤ 0.050.

## 3. Results

Frequencies and percentages of WHO weight status categories are indicated in Figure 1. No statistically significant differences were noted with respect to sex, obtaining a similar distribution between boys and girls. Overall, 28.0% of the children were overweight (*n* = 104), while 24.3% had obesity (*n* = 90). Furthermore, 52.4% of the pupils presented excess weight (*n* = 194).

Data of age, anthropometric characteristics, PF tests, screen time, sport activities hours, and Test Krece Plus score of the sample in accordance with different weight status categories (normal weight, overweight, and obesity) and sex are shown in Table 1. For boys, normal weight children showed higher CRF, relative handgrip strength, lower muscular strength, speed-agility, and sport activity hours than overweight or obese children. Consequently, post hoc assessments for PF tests revealed significant differences between normal weight participants and the two groups with excess weight; however, only statistically significant differences (*p* < 0.001) between obesity and normal weight peers were found. In the case of girls, the normal weight group had higher CRF, relative handgrip strength, lower muscular strength, sport activity hours, as well as PA level (assessed by Krece Plus Test). Furthermore, post hoc comparisons only showed statistically significant differences among the normal weight group and the two groups with excess weight for relative handgrip strength. Notwithstanding, post hoc analyses only indicated statistically significant differences between the normal weight and obesity groups for CRF, relative handgrip strength, sports activity hours, and PA level (assessed by Krece Plus Test).

On the other hand, Table 2 indicates the different correlations (bivariate and partials) found according to the BMI, WC, %BF, and the variables related to both PF and PA. A negative statistically significant correlation was shown between the BMI and PF tests; moderate for CRF (*rho* = −0.389) and standing broad jump, (*rho* = −0.340) and strong for the relative handgrip strength (*rho* = −0.547). At the same time, a positive statistically significant relationship between BMI and the time spent in 4 × 10 m Shuttle Run Test (*rho* = 0.263) was shown. Regarding PA, a negative statistically significant correlation was observed between BMI and sport activity hours (*rho* = −0.178) and the Krece Plus Test score (*rho* = −0.141); both correlations being very low. Regarding WC and %BF, relationships similar to those found for BMI and the different variables of PF, PA, and SB were identified.

Lastly, Table 3 shows the probability of obtaining different results in continuous variables according to the category of weight status. The obesity group was established as a reference category. Thus, compared to the obesity group, in the normal weight group, a greater probability of having a higher CRF (OR = 1.58; CI_95%_ = 1.38–1.82) was observed. In the case of handgrip strength, normal weight group had a lower probability of having higher handgrip strength (OR = 0.73; CI_95%_ = 0.65–0.82); that not being the case for relative handgrip strength (OR = 1.25; CI_95%_ = 1.19–1.31). Similarly, normal weight participants had a lower probability of spending more time in the 4x10m Shuttle Run Test (OR = 0.57; CI_95%_ = 0.45–0.72) than the obesity group. Regarding PA patterns, the normal weight group presented higher possibilities of doing more hours of sport activities (OR = 1.40; CI_95%_ = 1.19–1.66), and a higher score of PA level (assessed by Krece Plus Test) (OR = 1.23; CI_95%_ = 1.07–1.42) than their obese counterparts.

## 4. Discussion

The aim of this study was to describe, examine, and compare the level of PF and PA of pupils aged 6–13 in the Region of Murcia, related to sex and weight status. The results suggest that obese and overweight participants had lower results for health-related PF, sport activity hours, and PA level (measured by Krece Plus Short Test); this was not the case for screen time. Likewise, there were negative relationships between all anthropometric variables (WC, BMI, %BF) and Krece Plus Short Test, sport activities hours, CRF, upper, and lower muscular fitness and speed-agility. Similarly, normal weight participants were more likely to have superior results on PF tests, as well as more sport activity hours than obese participants.

Taking WHO cut-off points for weight status into consideration, in all the sample analysed, it was found that one out of five children had excess weight, obtaining very close values between boys and girls. If we consider the results published in the last ALADINO study in Spain [2], the data that we obtained in our analysis slightly exceeds those, which is in line with the alerts for excess weight indicated by the WHO [1]. 

The results of the PF tests showed that obese subjects presented lower values than normal weight participants on tests that required acceleration of body mass. Hence, obese subjects had lower values of CRF compared to the other groups (in both girls and boys), which is similar to the findings described in former studies [22,39,40,41]. Similarly, these results were comparable to other studies conducted in Spain [22,42,43], as well as in the Region of Murcia [25,26]. The results were not surprising due to the fact that decrease in functional residual capacity is the most frequent pulmonary function anomaly in obese children and adolescents [44]. On the contrary, a higher score of overweight and obese children was presented in the handgrip strength test, which does not involve a weight-bearing activity and does not consider the body mass of the participants. One explanation for this might be their increased fat-free mass, given that overweight and obese subjects seemed to have more body fat and fat-free mass than the normal weight group [45]. However, regarding relative muscle strength, obese participants presented inferior results than normal weight children, in both sexes. Previous studies in handgrip strength test revealed similar results to ours [39,46,47]. Similarly, lower results were found in both obese boys and girls for the standing broad jump test, as seen in other studies [39,46,47]. The difficulty obese children face in carrying their extra body weight in tasks involving propulsion or lifting their body [47] could be the reason for minor power performances observed in this study. Likewise, this condition also appears to affect speed-agility, since subjects with normal weight presented better results than the obese ones, as was also described in previous studies; nevertheless, no significant statistical differences were found in the case of girls, although the obese ones were slower than the other groups. Generally, girls present an earlier maturation process, two years approximately prior to boys [48], which affects variability and speed of adolescent growth in PF stability [49].

In addition, the Krece Plus Short Test offered higher scores for both boys and girls (only statistically significant differences for girls) for participants with normal weight (the same between normal and overweight in the case of girls). These results agree with those attained by Serra et al. [20] in the EnKid study, where it was corroborated that the chances of being overweight and obese was lower in children and adolescents who played sports, compared to those who led a more sedentary lifestyle. This could be explained by the lower level of motor competence in overweight/obese children; it is crucial to focus on motor skill improvement to promote regular participation in PA, especially, if they are not playing sports in a club environment [50]. Furthermore, these findings are similar to those obtained in other studies [24,51]. Consequently, obese children showed lower levels of PA. The same occurs in other countries, as is shown in studies that have found that a lack of PA increases the risk of becoming overweight or obese [52]. Besides, although there is no causal association between not being active and being overweight, it seems to be a behavioural risk factor of high importance, among others, and is associated with a sedentary lifestyle (hours spent watching TV and lack of sleep) [53].

As a consequence, obesity in youth and the rise in prevalence of excess weight is also related to the increasing amount of screen time [14]. Nevertheless, contrary to the findings of other authors [24,54], no statistically significant differences were found for screen time with respect to weight status in our sample. Notwithstanding the above, among the three weight status groups in our study, we found a high number of screen time hours, justified by the excessive time spent on this activity by Spanish children. It was recently found that more than half of Spanish children and adolescents do not comply with the international recommendations on the use of screens during the week proposed by Tremblay et al. [19]; and almost 80% do not comply with them on the weekend [32]. 

Our results advocate the need for an increase in both CRF and muscular fitness in children with obesity in order to foster autonomy and engagement in physical activity and programmes designed to raise quality of life [12]. Moreover, encouraging children to achieve and maintain high levels of CRF is highly recommended, as it is a modifiable indicator of long-term results [55]. Similarly, the effectiveness of exercise interventions aimed at gross motor skills, combined or not with another intervention (parental information, nutrition, etc.) so as to lessen weight-related consequences such as body mass index, body fat, and waist circumference, was recently reported [56].

This study has certain limitations. One the one hand, this research had a cross-sectional design. For this reason, it is not possible to conclude that the observed associations reveal causal relationships. Another limitation relates to the lack of information on the physical development of children. For instance, Tanner Stage, as a measure of maturity stage, could have offered a superior adjusted figure of CRF, since evidence of its usage to control for fitness testing has been found [57]. Lastly, if we had used accelerometer devices in our standardized questionnaires, it would have given us a more accurate calculation of PA patters and SB.

Regarding the strengths that characterise our study, even though BMI is generally used to identify overweight and obesity among children, we examined several anthropometric parameters, such as WHtR, WC, or %BF, giving us a better understanding of childhood obesity. This choice was justified on the basis of previously established recommendations, which indicate that BMI assessment should be accompanied by indices of body shape (WC, for instance) and other aspects (blood pressure, and cholesterol and glucose levels) to effectively determine individual risk [58]. Similarly, future studies could be analysed by utilising biomarkers to show further relations between health-related PF and cardiovascular risk in overweight and obese children, although in this age group, it is difficult to find altered biochemical parameters [59].

## 5. Conclusions

When comparing weight status, children that presented normal weight achieved higher results in health-related PF and PA than children in the other groups (overweight and obese). However, this did not occur in the case of the handgrip strength test, due to the fact that the body weight of the participants was not considered. In terms of adiposity and WC, a negative relation with PF and PA was also verified. Nevertheless, for SB, this was not found to be the case with respect to the anthropometric parameters.

The authors emphasise the need for changes in public policies and school-based intervention programmes to develop higher levels of both PF and PA in overweight and obese children so as to counteract the adverse health effects caused by excess weight. Moreover, these results can be implemented in planning structured and tailored intervention programs to prevent overweight issues and obesity in children. These intervention programs should also focus on motor skill improvement, particularly when children do not practise sports in a club environment, because if they do not participate in directed or regulated activities, it will be more difficult for them to engage in PA on a regular basis.

## Figures and Tables

**Figure 1 ijerph-17-04518-f001:**
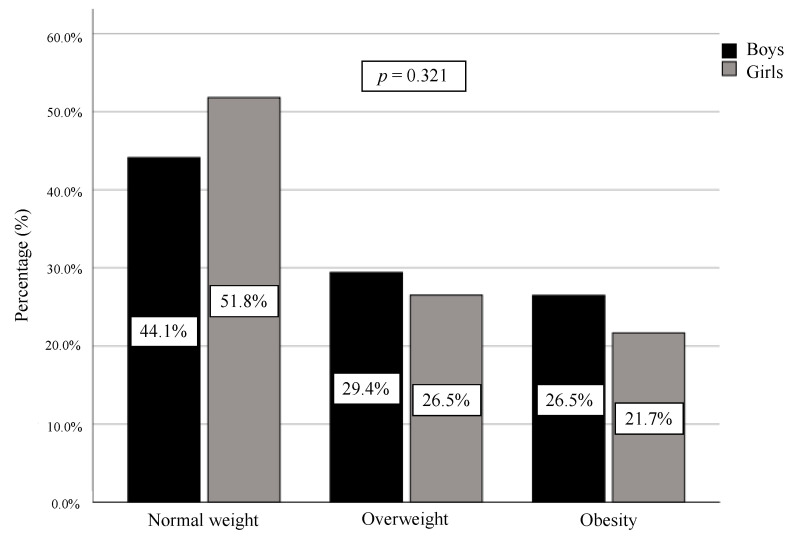
Weight status in respect to sex according to World Health Organization growth references in pupils aged 6 to 13 in the Region of Murcia (Spain).

**Table 1 ijerph-17-04518-t001:** Physical fitness, physical activity levels, and sedentary behaviour according to weight status in pupils aged 6–13.

Variables	Boys					Girls				
	Normal Weight *n* = 90 (44.1%)	Overweight *n* = 60 (29.4%)	Obesity *n* = 54 (26.5%)	*p*	*d*	Normal Weight *n* = 86 (51.8%)	Overweight *n* = 44 (26.5%)	Obesity *n* = 36 (21.7%)	*p*	*d*
Age (years)	8.8 ± 1.7	8.9 ± 1.8	8.7 ± 1.8	0.788	0.18	8.5 ± 1.8	8.6 ± 1.9	8.4 ± 1.7	0.909	0.21
Weight (kg)	30.6 ± 6.5	36.5 ± 7.7 ^a^	46.1 ± 12.3 ^a,b^	<0.001 **	1.44	28.8 ± 7.0	37.1 ± 9.7 ^a^	46.5 ± 10.6 ^a,b^	<0.001 **	1.55
Height (m)	1.36 ± 0.12	1.37 ± 0.10	1.38 ± 0.11	0.110	0.23	1.32 ± 0.12	1.36 ± 0.14	1.36 ± 0.10	0.081	0.28
BMI (kg/m^2^)	16.34 ± 1.12	19.23 ± 1.47 ^a^	23.47 ± 3.34 ^a,b^	<0.001 **	3.58	16.30 ± 1.36	19.67 ± 1.55 ^a^	24.73 ± 3.62 ^a,b^	<0.001 **	3.47
BMI (z-score)	0.10 ± 0.57	1.50 ± 0.28 ^a^	2.81 ± 0.62 ^a,b^	<0.001 **	4.38	0.88 ± 0.62	1.49 ± 0.29 ^a^	2.77 ± 0.64 ^a,b^	<0.001 **	3.94
WC (cm)	58.7 ± 4.1	62.6 ± 5.1 ^a^	70.9 ± 8.6 ^a,b^	<0.001 **	1.65	56.01 ± 6.33	61.72 ± 4.87 ^a^	71.26 ± 7.39 ^a,b^	<0.001 **	1.92
WHtR (WC (cm)/Height (cm))	0.43 ± 0.02	0.46 ± 0.02 ^a^	0.51 ± 0.04 ^a,b^	<0.001 **	2.20	0.43 ± 0.04	0.46 ± 0.03 ^a^	0.52 ± 0.05 ^a,b^	<0.001 **	2.10
BF (kg)	6.2 ± 2.6	10.0 ± 4.1 ^a^	17.2 ± 8.0 ^a,b^	<0.001 **	1.89	6.7 ± 2.5	10.8 ± 4.1 ^a^	18.1 ± 5.8 ^a,b^	<0.001 **	2.06
BF (%)	19.6 ± 5.5	26.7 ± 6.8 ^a^	35.7 ± 7.7 ^a,b^	<0.001 **	2.18	22.8 ± 4.3	28.6 ± 5.4 ^a^	38.1 ± 5.2 ^a,b^	<0.001 **	2.62
FFM (kg)	24.4 ± 4.7	26.4 ± 4.7	28.9 ± 5.3 ^a,b^	<0.001 **	0.98	22.1 ± 4.8	26.3 ± 6.4	28.5 ± 5.3 ^a,b^	<0.001 **	0.99
FFM (%)	80.4 ± 5.5	73.3 ± 6.8 ^a^	64.3 ± 7.7 ^a,b^	<0.001 **	2.18	77.2 ± 4.3	71.4 ± 5.4 ^a^	61.9 ± 5.2 ^a,b^	<0.001 **	2.62
Handgrip strength (kg)	12.98 ± 3.58	13.95 ± 4.18	14.25 ± 4.16	0.124	0.28	11.25 ± 3.40	13.01 ± 4.18	13.60 ± 3.64 ^a^	0.002 *	0.55
Handgrip strength/BW	0.42 ± 0.07	0.38 ± 0.08 ^a^	0.31 ± 0.07 ^a,b^	<0.001 **	0.63	0.39 ± 0.07	0.35 ± 0.06 ^a^	0.29 ± 0.05 ^a,b^	<0.001 **	1.19
Standing broad jump (cm)	124.5 ± 26.6	118.9 ± 23.3	104.9 ± 21.7 ^a,b^	<0.001 **	1.19	112.5 ± 23.4	107.2 ± 21.9	98.3 ± 19.6 ^a^	<0.001 **	0.50
4 × 10 m Shuttle Run Test (s)	13.15 ± 1.38	13.51 ± 1.42	14.19 ± 1.65 ^a^	<0.001 **	0.55	13.95 ± 1.21	13.97 ± 1.33	14.48 ± 1.37	0.151	0.21
20 m Shuttle Run Test (laps)	25.7 ± 15.0	22.7 ± 13.3	11.9 ± 7.0 ^a,b^	<0.001 **	0.93	19.3 ± 10.5	16.2 ± 8.3	10.7 ± 3.4 ^a,b^	<0.001 **	0.81
CRF (mL/kg/min)	46.47 ± 4.31	45.52 ± 4.42	43.25 ± 3.76 ^a,b^	<0.001 **	0.61	45.50 ± 3.38	44.29 ± 4.23	42.91 ± 3.24 ^a^	0.004 *	0.49
Daily screen time (hours)	3.3 ± 1.0	3.2 ± 1.2	3.5 ± 0.9	0.658	0.15	3.3 ± 1.0	3.5 ± 0.7	3.4 ± 0.9	0.584	0.15
Weekly sport activities (hours)	3.2 ± 1.6	2.8 ± 1.8	2.4 ± 1.7 ^a^	0.030 *	0.32	2.8 ± 1.5	2.6 ± 1.6	1.8 ± 1.3 ^a^	0.005 *	0.48
Krece Plus Short Test (score)	6.5 ± 1.9	6.1 ± 2.1	5.8 ± 2.0	0.107	0.22	6.1 ± 1.8	6.1 ± 1.8	5.2 ± 1.6 ^a^	0.017 *	0.40

Note: Data expressed as Mean ± SD. BF—Body fat; BMI: Body mass index; BW—Body weight; CRF—Cardiorespiratory fitness; FFM—Free-fat mass; WC—Waist circumference; WHtR—Waist-to-height ratio. ^a^—Statistically significant differences from normal weight (*p* ≤ 0.050). ^b^—Statistically significant differences from overweight (*p* ≤ 0.050). *d*—Effect size (Cohen’s *d*). *—*p* ≤ 0.050; **—*p* ≤ 0.001.

**Table 2 ijerph-17-04518-t002:** Bivariate and partial correlations between different anthropometric measures and physical fitness, physical activity, and sedentary behaviour variables.

Variables	BMI		WC		%BF	
	Crude	Adjusted ^#^	Crude	Adjusted ^#^	Crude	Adjusted ^#^
Handgrip strength (kg)	0.427 **	0.348 **	0.632 **	0.416 **	0.350 **	0.228 **
Handgrip strength/BW	−0.482 *	−0.547 **	−0.291 **	−0.432 **	−0.462 **	−0.535 **
Standing broad jump (cm)	−0.127 *	−0.340 **	0.060	−0.298 **	−0.232 **	−0.572 **
4 × 10 m Shuttle Run Test (s)	0.038	0.263 **	−0.153 *	0.214 **	0.166 **	0.401 **
20 m Shuttle Run Test (laps)	−0.295 **	−0.394 **	−0.165 **	−0.338 **	−0.363 **	−0.469 **
CRF (mL/kg/min)	−0.462 **	−0.389 **	−0.536 **	−0.336 **	−0.537 **	−0.455 **
Daily screen time (hours)	0.037	0.025	0.030	0.008	0.026	0.017
Weekly sport activities (hours)	−0.154 *	−0.178 *	−0.082	−0.149 *	−0.157 *	−0.182 **
Krece Plus Short Test (score)	−0.118 *	−0.141 *	−0.059	−0.126 *	−0.134 **	−0.149 **

Note: BF—Body fat; BMI—Body mass index; BW—Body weight; CRF—Cardiorespiratory fitness; WC—Waist circumference. ^#^—Adjusted by sex and age. *—*p* ≤ 0.050; **—*p* ≤ 0.001.

**Table 3 ijerph-17-04518-t003:** Multinomial logistic regression for continuous variables of study related to weight status.

Predictors	Normal Weight	Overweight	Obesity
Model 1			
Handgrip strength (per 1 kg)	0.88 ** (0.83–0.94)	0.97 (0.91–1.04)	1
Handgrip strength/BW (per 0.01 unit)	1.23 ** (1.17–1.28)	1.14 ** (1.09–1.19)	1
Standing broad jump (per 1 cm)	1.03 ** (1.02–1.04)	1.01 ** (1.01–1.03)	1
4 × 10m Shuttle Run Test (per 1 s)	0.70 ** (0.58–0.84)	0.76 * (0.63–0.93)	1
20m Shuttle Run Test (per 1 lap)	1.14 ** (1.10–1.19)	1.13 ** (1.08–1.18)	1
CRF (ml/kg/min) (per 1 unit)	1.19 ** (1.12–1.28)	1.13 ** (1.05–1.21)	1
Daily screen time (per 1 h)	1.15 (0.88–1.50)	1.10 (0.82–1.47)	1
Weekly sport activities (per 1 h)	1.37 ** (1.17–1.62)	1.23 * (1.03–1.47)	1
Krece Plus Short Test (per 1 unit)	1.22 * (1.06–1.40)	1.13 (0.98–1.32)	1
Model 2			
Handgrip strength (per 1 kg)	0.73 ** (0.65–0.82)	0.90 (0.81–1.01)	1
Handgrip strength/BW (per 0.01 unit)	1.25 ** (1.19–1.31)	1.15 ** (1.09–1.20)	1
Standing broad jump (per 1 cm)	1.04 ** (1.03–1.06)	1.03 ** (1.01–1.04)	1
4x10m Shuttle Run Test (per 1 s)	0.57 ** (0.45–0.72)	0.72 * (0.56–0.91)	1
20m Shuttle Run Test (per 1 lap)	1.15 ** (1.11–1.20)	1.13 ** (1.08–1.18)	1
CRF (ml/kg/min) (per 1 unit)	1.58 ** (1.38–1.82)	1.44 ** (1.25–1.66)	1
Daily screen time (per 1 h)	0.86 (0.66–1.13)	0.91 (0.68–1.21)	1
Weekly sport activities (per 1 h)	1.40 ** (1.19–1.66)	1.23 * (1.04–1.47)	1
Krece Plus Short Test (per 1 unit)	1.23 * (1.07–1.42)	1.13 (0.97–1.32)	1

Note: Data expressed as OR (CI_95%_). BW—Body weight; CRF—Cardiorespiratory fitness. Obesity group selected as the reference category. Model 1: Unadjusted; Model 2: Adjusted by sex and age. *—*p* ≤ 0.050; **—*p* ≤ 0.001.

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
