# Peer review of "Weight Status Is Related to Health-Related Physical Fitness and Physical Activity but Not to Sedentary Behaviour in Children"

_ijerph, 2020, doi:10.3390/ijerph17124518_

Round 1
Reviewer 1 Report
Remarks to author-
- Whole research article is focused on relationship between body weight/body weight compositions NOT body compositions with physical fitness and physical activity . Hence author should revise the title of the manuscript and write body weight composition instead of body composition.
- If author wants to focus on body composition, then author should at least evaluate body fat composition and add that data in these subjects based on the data available.
- Write full form of CRF in line no.68.
- Add some introduction regarding relationship between CVD and obesity (body weight).
- Rephrase line 100 "in which took part a total of 370 children" .
- Full form of WHO line 117
Author Response
Reviewer 1
Thanks for your valuable comments. We have tried to address all issues mentioned and we have answered each of them.
- Whole research article is focused on relationship between body weight/body weight compositions NOT body compositions with physical fitness and physical activity. Hence author should revise the title of the manuscript and write body weight composition instead of body composition.
Thanks for this advice. We have been agreeing on the term to be used. We finally opted for the term "weight status". We do not use the term "composition", because we have not included other body composition parameters such as bone mass. This choice is based on a previous search of the scientific literature.
- If author wants to focus on body composition, then author should at least evaluate body fat composition and add that data in these subjects based on the data available.
Yes, we consider that it is better, thanks for the contribution. As we said in the previous comment, we have modified the term “body composition”.
- Write full form of CRF in line no.68.
Thanks for the observation. It was a mistake.
- Add some introduction regarding relationship between CVD and obesity (body weight).
We have added further information as follows: “and has a great influence on risk factors for cardiovascular disease and the development of atherosclerosis (Bridger, 2009)”. Likewise, we have specified examples of these cardiovascular risk parameters (i. e., cholesterol, triglycerides, blood pressure level, etc.), to a better understanding.
Bridger, T. (2009). Childhood obesity and cardiovascular disease. Paediatrics & Child Health, 14(3), 177–182. https://doi.org/10.1093/pch/14.3.177
- Rephrase line 100 "in which took part a total of 370 children".
We have clarified this part of the text as follows: “being included a total of 370 children”.
- Full form of WHO line 117
Thanks for the observation. It was modified.
Reviewer 2 Report
The topic of relationships between body build and body composition of children with several elements of their lifestyle is interesting. However, the presented manuscript requires some changes.
- The biggest doubts are related to the analyzed group, which includes children aged 6-13 years. The authors of the manuscript wrote in the discussion chapter, that children in the studied group were at various stages of development and puberty. And these are the processes affecting the body structure and composition. Perhaps it would be better to separate age groups, e.g. : childhood (including boys or girls aged 6 years), juvenile (including boys aged 7-12 years and girls aged 7-10 years), adolescent (including boys aged over 12 years and girls aged over 10 years) (Bogin B &Varia C (2017) Evolution of human life history. In Evolution of nervous systems 2e vol.4 pp37-50).
- The size of the group seems to be insufficient. The Authors of the manuscript should also present a table with the number of participanta in particular age groups. The lack of such information raises doubts as to whether the group of boys and girls was comparable. Did one of the groups dominate, e.g. children, and in the other e.g. adolescence? .
- p. 3, line 114 I would replace “electronic balance” by “body composition analyser”.
- p. 3, line 119 I would replace “navel” by „umbilicus”
- In the title I would replace world “pupils” with the word “children” .
- I would also like to ask to explain why the Authors of manuscript calculated the body fat percentage (BF%) using the Siri formula, since they used body composition analyser - Tanita?
Best regards!
Author Response
Reviewer 2
Thanks for your valuable comments. We agree and we consider too important to describe the situation in different regions with specific features due to we have already global and country information about this matter.
The topic of relationships between body build and body composition of children with several elements of their lifestyle is interesting. However, the presented manuscript requires some changes.
- The biggest doubts are related to the analyzed group, which includes children aged 6-13 years. The authors of the manuscript wrote in the discussion chapter, that children in the studied group were at various stages of development and puberty. And these are the processes affecting the body structure and composition. Perhaps it would be better to separate age groups, e.g. : childhood (including boys or girls aged 6 years), juvenile (including boys aged 7-12 years and girls aged 7-10 years), adolescent (including boys aged over 12 years and girls aged over 10 years) (Bogin B &Varia C (2017) Evolution of human life history. In Evolution of nervous systems 2e vol.4 pp37-50).
Yes, we consider that it is better, thanks for the contribution. However, our sample is not large enough to segment by age. To deal with this, correlation and regression analyses were adjusted for age and sex, as can be seen in other previously published studies. We must not forget that the aim of our study was “to describe, compare and analyse the level of PF, PA and SB in children aged 6-13 in the Region of Murcia”; not with not having as a main objective the stratification by age.
In addition, we will take this aspect into account, as well as the indicated reference for future studies.
- The size of the group seems to be insufficient. The Authors of the manuscript should also present a table with the number of participanta in particular age groups. The lack of such information raises doubts as to whether the group of boys and girls was comparable. Did one of the groups dominate, e.g. children, and in the other e.g. adolescence?.
In addition to a previous answer, we have included a division of the different age group for a better understanding in the “Study design and participants” statement. We included the following statement: “In terms of age groups, 230 schoolchildren were in the range of 6-9 (118 boys and 112 girls). For the range 10-13, 140 schoolchildren were found (86 boys and 54 girls)”. If it is not clear, please do not hesitate to indicate to us.
- p. 3, line 114 I would replace “electronic balance” by “body composition analyser”.
Thanks for the observation. It was modified.
- p. 3, line 119 I would replace “navel” by „umbilicus”
Thanks for the observation. It was a translation mistake. We have modified as follows: “at the intersection between the border of the iliac crest and the last rib”.
- In the title I would replace world “pupils” with the word “children”.
Thanks for the contribution. It was modified.
- I would also like to ask to explain why the Authors of manuscript calculated the body fat percentage (BF%) using the Siri formula, since they used body composition analyser - Tanita?
Thank you for your valuable feedback. We only use the Tanita in order to obtain the children’s weight. We could indicate the body composition from Tanita, but we decided to perform skinfolds measurements by trained and accredited staff by the International Society for the Advancement of Kinanthropometry (ISAK). Moreover, skinfold thickness and bioimpedance analysis should not be used interchangeably in children and adolescents (Forte et al., 2020).
On the other hand, the fact of providing a lot of anthropometric parameters (apart from BMI) is one of the strengths of our study, as previously recommended (González-Muniesa et al., 2017).
Forte, G. C., Rodrigues, C. A. S., Mundstock, E., Santos, T. S. dos, Filho, A. D., Noal, J., Amaral, M. A., Preto, L. T., Vendrusculo, F. M., & Mattiello, R. (2020). Can skinfold thickness equations be substituted for bioimpedance analysis in children? Jornal de Pediatria, S0021755719306023. https://doi.org/10.1016/j.jped.2019.12.006
González-Muniesa, P., Mártinez-González, M.-A., Hu, F. B., Després, J.-P., Matsuzawa, Y., Loos, R. J. F., Moreno, L. A., Bray, G. A., & Martinez, J. A. (2017). Obesity. Nature Reviews Disease Primers, 3(1), 17034. https://doi.org/10.1038/nrdp.2017.34
Best regards!
Thanks for your comments and your valuable labour reviewing our manuscript. We hope to have addressed all the specific comments you have provided us.